# A high-throughput small molecule screen identifies farrerol as a potentiator of CRISPR/Cas9-mediated genome editing

**Weina Zhang[1†], Yu Chen[1†], Jiaqing Yang[1], Jing Zhang[1], Jiayu Yu[1], Mengting Wang[1], Xiaodong Zhao[1], Ke Wei[1], Xiaoping Wan[1], Xiaojun Xu[2], Ying Jiang[1], Jiayu Chen[1]\*, Shaorong Gao[1]\*, Zhiyong Mao[1,3]\***

[1]Clinical and Translational Research Center of Shanghai First Maternity & Infant Hospital, Shanghai Key Laboratory of Signaling and Disease Research, Frontier Science Center for Stem Cell Research, School of Life Sciences and Technology, Tongji University, Shanghai, China; [2]State Key Laboratory of Natural Medicines, China Pharmaceutical University, Nanjing, China; [3]Tsingtao Advanced Research Institute, Tongji University, Qingdao, China

**\*For correspondence:**
chenjiayu@tongji.edu.cn (JC);
gaoshaorong@tongji.edu.cn (SG);
zhiyong_mao@tongji.edu.cn (ZM)

[†]These authors contributed equally to this work

**Competing interests:** The authors declare that no competing interests exist.

**Abstract** Directly modulating the choice between homologous recombination (HR) and non-homologous end joining (NHEJ) - two independent pathways for repairing DNA double-strand breaks (DSBs) - has the potential to improve the efficiency of gene targeting by CRISPR/Cas9. Here, we have developed a rapid and easy-to-score screening approach for identifying small molecules that affect the choice between the two DSB repair pathways. Using this tool, we identified a small molecule, farrerol, that promotes HR but does not affect NHEJ. Further mechanistic studies indicate that farrerol functions through stimulating the recruitment of RAD51 to DSB sites. Importantly, we demonstrated that farrerol effectively promotes precise targeted integration in human cells, mouse cells and mouse embryos at multiple genomic loci. In addition, treating cells with farrerol did not have any obvious negative effect on genomic stability. Moreover, farrerol significantly improved the knock-in efficiency in blastocysts, and the subsequently generated knock-in mice retained the capacity for germline transmission.

## Introduction

The ability to precisely edit genomes holds great promise for a wide range of applications in both biomedical research and the treatment of human genetic diseases (*Su et al., 2016*; *Tebas et al., 2014*). In recent years, a number of genome editing tools including zinc-finger nucleases (*Brunet et al., 2009*; *Miller et al., 2007*; *Porteus and Baltimore, 2003*), transcription activator-like effector nucleases (TALENs) (*Wood et al., 2011*) and the RNA-guided CRISPR/Cas9 nuclease system (*Mali et al., 2013*) have been developed. Among them, the adapted *Streptococcus pyogenes* CRISPR/Cas9 (SpCRISPR/Cas9) has received the greatest attention due to its simplicity, relative high precision and flexibility (*Jinek et al., 2012*). The SpCRISPR/Cas9-mediated genome editing system consists of the Cas9 nuclease protein and a single guide RNA (sgRNA) containing a 20-nucleotide (nt) sequence with complementary pairing to a target genomic locus adjacent to a 5'NGG3' proto-spacer adjacent motif (PAM). When combined with an sgRNA, the Cas9 nuclease generates a DNA double-strand break (DSB) around 3 bp upstream the target PAM sequence (*Cong et al., 2013*; *Mali et al., 2013*).

Upon DSB induction, two different DSB repair mechanisms are available to repair the lesion – homologous recombination (HR) and non-homologous end joining (NHEJ). The choice of DSB repair pathway determines the outcome of the genome editing. In the presence of a homologous

template, successful HR results in a precise knock-in event (*Ran et al., 2013*). By contrast, the error-prone NHEJ probably leads to a phenotype of gene knock-out (*Zhang et al., 2014*). Many factors, including cell cycle stage (*Yang et al., 2016*), competition between DNA damage repair factors such as RIF1/53BP1 vs. BRCA1/CtIP (*Hollick et al., 2003*; *Srivastava et al., 2012*) and cell type collectively influence the choice between HR and NHEJ. Great efforts have been made to improve the efficiency of SpCRISPR/Cas9-mediated knock-in (*Smirnikhina et al., 2019*). Knocking down the DNA damage response factor, 53BP1, which favors the choice of NHEJ (*Callen et al., 2013*); or important NHEJ factors such as KU70, KU80 and LIG4 (*Chu et al., 2015*) promotes the SpCRISPR/Cas9-mediated knock-in efficiency (*Ye et al., 2018*). In addition, a number of small molecules inhibiting NHEJ or promoting HR have been shown to improve knock-in efficiency (*Riesenberg and Maricic, 2018*). For instance, suppressing NHEJ by blocking LIG4 activity with SCR7, or inhibiting DNA-PKcs kinase activity with NU7441 or NU7026, has been shown to improve the precise targeting efficiency of SpCRISPR/Cas9 (*Chu et al., 2015*; *Robert et al., 2015*; *Zhang et al., 2017*). Similarly, stimulating the HR factor, RAD51, with RS-1 also improved SpCRISPR/Cas9 editing efficiency (*Jayathilaka et al., 2008*). However, both inhibiting NHEJ and stimulating the activity of the recombinase involved in HR are potentially harmful to the maintenance of genome integrity (*Chen et al., 2008*; *Vartak and Raghavan, 2015*). NHEJ is the major pathway for mending the broken ends in mammalian cells in all cell cycle stages (*Mao et al., 2008a*). Loss of this pathway often leads to high cancer incidences and premature aging (*Lombard et al., 2005*; *Vogel et al., 1999*). The risk of activating RAD51 is that it might increase the chance of the spontaneous recombination with prevalent repetitive sequences in mammalian cells, resulting in the loss of large amounts of genetic information (*Klein, 2008*; *Richardson et al., 2004*). Therefore, there is a need to expand the list of the compounds which can improve the efficiency of precise genome editing with minimal or no effect on global genome stability.

Here, based on our recently developed cell lines containing a dual-reporter for the simultaneous measurement of both HR and NHEJ efficiency at the same chromosomal site (*Chen et al., 2019*), we generated a novel, rapid, quantitative and easy-to-score screening platform for identifying compounds that could be potentially applied in SpCRISPR/Cas9-mediated genome editing. Using the system, we screened 722 small molecules isolated from herbs used in traditional Chinese medicine. We found that farrerol, a natural compound isolated from *Rhododendron dauricum*, which exhibits antibechic, anti-bacterial and anti-inflammatory properties (*Zhang et al., 2015*), promoted HR but had no effect on NHEJ. Further mechanistic studies indicated that farrerol stimulated the recruitment of RAD51 to DSB sites rather than changed the expression levels of HR related factors. We then demonstrated that farrerol facilitated SpCRISPR/Cas9-mediated genome targeting at different loci in human somatic cells, mouse embryonic stem cells (ESCs) and mouse embryos. In contrast, SCR7 and RS-1, two most commonly used small molecules for gene targeting, exhibited a position dependent effect on targeting efficiency in different cell lines and also in mouse embryos. Moreover, we demonstrated that farrerol did not destabilize genomes using comet assay, karyotype analysis and immunofluorescence staining of genotoxic markers. Conversely, treating cells with SCR7, which blocks the NHEJ pathway, and RS-1, which activates RAD51, diminished genome integrity. Besides, we also found that farrerol could significantly suppress the efficiency of single strand annealing (SSA), which may cause deletions of large fragments between repetitive sequences in genomes (*Wyman and Kanaar, 2006*), while RS-1 had no effect on SSA efficiency. Moreover, treating mouse ESCs and blastocysts with farrerol had no adverse effect on cell growth and embryo development, whereas we observed abnormal development in mouse blastocysts treated with SCR7, and we failed to observe a remarkable stimulatory effect on knock-in efficiency in RS-1 treated mouse blastocysts. Most intriguingly, farrerol significantly improved the efficiency of SpCRISPR/Cas9-mediated knock-in in blastocysts, and the generated knock-in founder mice retained the capability for germline transmission.

## Results

### Generation of a high-throughput compound screening platform for monitoring the choice of DSB repair pathway

To identify novel compounds that can potentially enhance precise genome editing efficiency, we created CLZ3, a cell line suitable for high-throughput drug screening. The cell line contains a reporter cassette for the simultaneous detection of both HR and NHEJ efficiency at the same locus upon DSB induction at the I-SceI recognition sites (*Figure 1A*), and a chromosomally integrated doxycycline-inducible vector encoding the endonuclease, I-SceI (*Figure 1B*). The CLZ3 cell line was derived from the previously described D4a human fibroblast cell line (*Chen et al., 2019*), which was developed to measure the efficiency of both HR and NHEJ simultaneously at the same chromosomal site. In the dual HR-NHEJ reporter cassette (*Figure 1A*), the part downstream of the CMV promoter consists of two GFP exons, the engineered rat Pem1 intron separating the two GFP exons, an adenoviral exon (AD2) flanked by recognition sequences for I-SceI endonuclease in an inverted orientation and an ATG-less tdTomato gene. In the part upstream of the CMV promoter, the reporter cassette contains the Pem1 intron and the inserted full length tdTomato gene. The inducible vector for I-SceI expression is comprised of two parts in an inverted orientation to avoid potential interference from one another (*Figure 1B*). The first part contains the coding sequence of reverse tetracycline-controlled transactivator (rtTA) driven by a constitutively active CAG promoter ($P_{CAG}$). The second part encodes the I-SceI-NLS (Nuclear localization sequence) -HA endonuclease driven by a TET response element (TRE) promoter.

In brief, in the absence of doxycycline, the CLZ3 cells do not express the I-SceI endonuclease, thus no DSBs are generated, and cells remain tdTomato and GFP negative. In contrast, supplementing CLZ3 cells with doxycycline turns on the I-SceI expression (*Figure 1C*), resulting in the induction of DSBs on the dual-reporter cassette. Successful repair by HR leads to functional tdTomato, therefore turning cells red, while NHEJ restores active GFP, turning cells green (*Figure 1A–B*). Indeed, using the CLZ3 cell line we found that adding doxycycline into the culture medium leads to the I-SceI expression (*Figure 1C*), and we observed both tdTomato$^+$ (4.3%) and GFP$^+$ cells (1.5%) either using confocal microscopy or FACS analysis (*Figure 1D–E*). These data indicate that we successfully obtained the cells for screening compounds that potentially alter the balance between HR and NHEJ.

### The identification of farrerol as an enhancer of HR but not NHEJ using the CLZ3 cell line

We then employed the CLZ3 cell line to screen a library containing 722 small molecules isolated from herbs used in traditional Chinese medicine. CLZ3 cells were seeded at a density of 10,000 cells per well into a 12-well plate. On day 1 post seeding, cells were supplemented with the chemical compounds at a concentration of 5 μM. On day 2, doxycycline was then added into the culture medium at a concentration of 0.5 μg/mL. On day 4 post splitting, cells were harvested for FACS analysis of DSB repair efficiency (*Figure 1F*). Among the 722 screened compounds, we found that it was farrerol that could significantly improve HR repair but had no detectable effect on NHEJ (*Figure 1G*). We further validated its stimulatory effect on HR using our well-established HCA2-H15c and HCA2-I9a cell lines for analyzing HR and NHEJ efficiency separately (*Mao et al., 2008a*). Consistent with our observation in CLZ3 cells, farrerol was proved to significantly promote HR and had no obvious influence on NHEJ in these cell lines (*Figure 1—figure supplement 1A*). In addition, using the extrachromosomal assay we also demonstrated that farrerol significantly promoted HR repair efficiency in mouse ESCs (*Figure 1—figure supplement 1B*).

To elucidate the molecular mechanisms of HR activation by farrerol, the protein levels of HR associated factors including RPA2, NBS1, RAD50, RAD51, MRE11, CtIP, BRCA1, EXO1 XRCC2, and 53BP1, a negative regulator of HR, were analyzed by western blot (*Figure 1—figure supplement 2A*). We did not observe any significant alterations in the expression level of repair-related factors upon farrerol treatment, indicating that enhancement of HR efficiency is probably not due to changes in the expression levels of the analyzed HR factors. Similarly, cell cycle distribution was not significantly impacted by farrerol treatment (*Figure 1—figure supplement 2B*), suggesting that the promotion of HR by farrerol is not through arresting cells in the HR-dominant S/G2 stage. We then

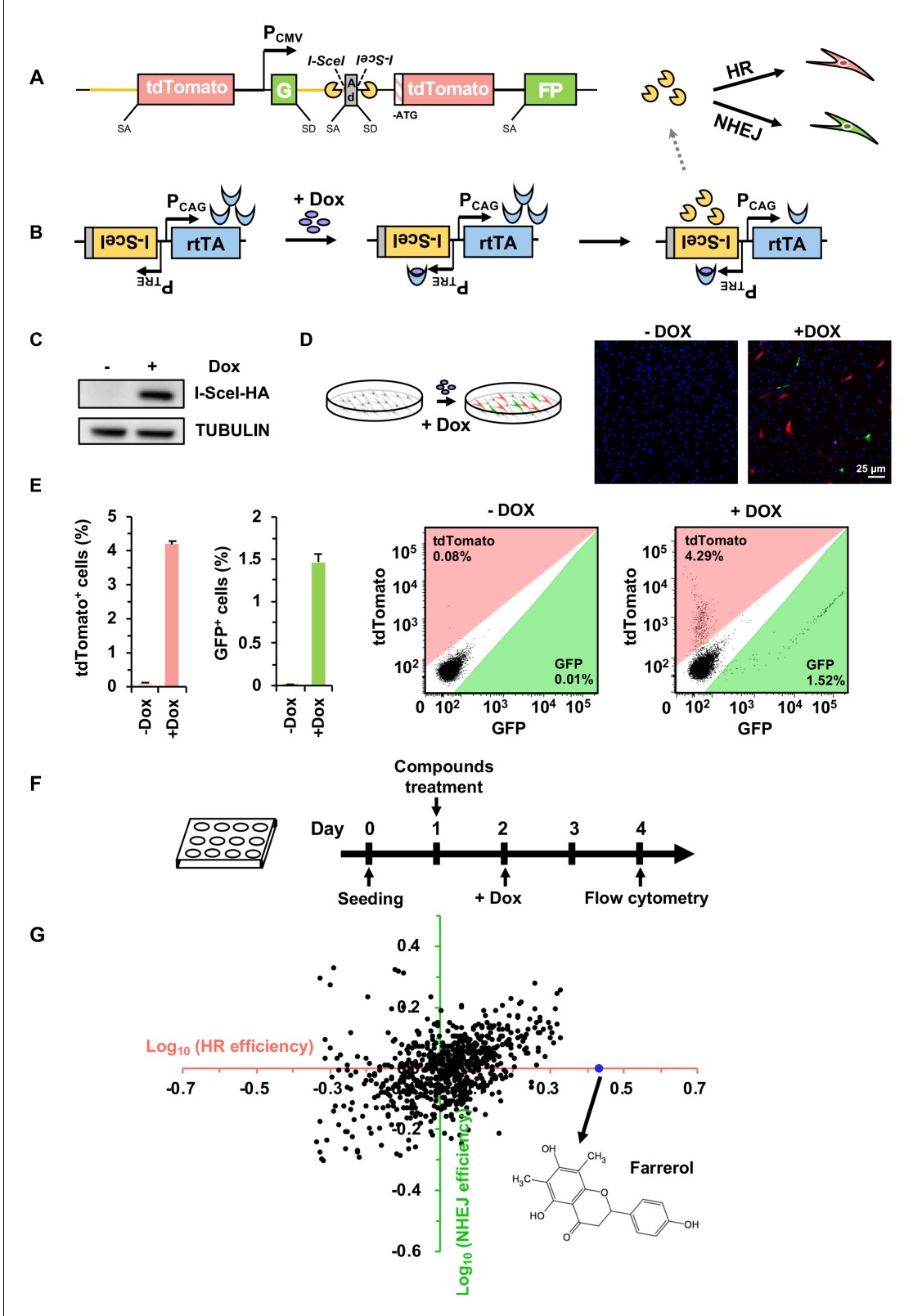

**Figure 1.** Establishment of a compound screening platform for monitoring the efficiency of DSB repair by HR and NHEJ. (**A**) Diagram of the HR-NHEJ dual fluorescent reporter (*Chen et al., 2019*). SD, splice donor; SA, splice acceptor. In the reporter cassette for simultaneous analysis of HR and NHEJ at the same chromosomal site, it contains two parts separated by a CMV promoter. The part downstream of the promoter contains two GFP exons separated by the engineered Pem1 intron with splice donor and acceptor, an adenoviral exon (AD2) with splice donor and acceptor, two I-SceI

*Figure 1 continued on next page*

*Figure 1 continued*

recognition sites in an inverted orientation, and an ATG-less tdTomato gene. The second part upstream of the promoter contains the Pem1 intron and a full-length tdTomato gene with a splice donor before start codon ATG. The homology region between the two parts of the reporter is approximately 3.9 kb including the rat Pem1 intron (marked in yellow line) and the tdTomato gene. (B) Schematic diagram of the doxycycline-inducible I-SceI expression vector. rtTA, reverse tetracycline trans-activator protein; PTRE, promoter of tetracycline response element; PCAG, the CAG promoter, a strong synthetic promoter frequently used to drive high levels of gene expression in mammalian systems. (C–D) Generation of the compound screening platform. The linearized doxycycline-inducible I-SceI expression vector was nucleofected into D4a cells followed by selection with 30 µg/mL hygromycin B. Then individual colonies were picked and one of the colonies, CLZ3, was utilized for further studies. I-SceI endonuclease expression could be efficiently induced upon doxycycline supplementation (C), further resulting in the generation of DSBs on the reporter cassette. Successful repair by HR or NHEJ results in functional tdTomato or GFP, turning cells red or green respectively, which could be observed via microscopy (D). (E) Flow cytometry analysis of tdTomato$^+$ and GFP$^+$ cells upon doxycycline addition (left panel). The representative FACS traces are shown (right panel). (F) Workflow of compound screening using the CLZ3 cell line. (G) 722 small molecules were screened using CLZ3 cells and farrerol (highlighted in blue) was identified as an HR enhancer.

The online version of this article includes the following source data and figure supplement(s) for figure 1:

**Source data 1.** The list of small molecule compounds and their influences on DNA repair.
**Source data 2.** Summary of NHEJ, HR and SSA efficiency.
**Figure supplement 1.** Farrerol stimulates the precise HR in human fibroblasts and mouse ESCs.
**Figure supplement 2.** Farrerol accelerates the recruitment of RAD51 rather than influences the expression of the indicated HR-related factors.
**Figure supplement 3.** Farrerol inhibits the SSA efficiency.

hypothesized that farrerol might improve HR by accelerating the recruitment of HR factors to DNA damage sites. Accordingly, we checked the recruitment of RPA2, a single-strand DNA binding protein, at indicated time points post 2 Gy ionizing irradiation (IR) using immunofluorescence assay (*Figure 1—figure supplement 2C–D*). However, the result indicated that farrerol has no influence on the recruitment of RPA2. Surprisingly, we found that the average foci number of RAD51 per nucleus was 6.5 and 6.1 in the presence of farrerol (1 and 5 µM) at 4 hr post IR at 2 Gy, in comparison to 4.4 in control group (*Figure 1—figure supplement 2E–F*). At 16 hr post IR, the average foci number of RAD51 per nucleus dropped to 3.2 and 4.3 in farrerol treating cells whereas there were still 6.4 RAD51 foci in the control group (*Figure 1—figure supplement 2E–F*). These data indicate that the rapid recruitment and timely clearance of RAD51 post IR may contribute to the promotion of HR by farrerol.

In mammalian cells HR can be further categorized into precise gene conversion, crossover, and the SSA pathway which results in deletions of DNA sequences between two direct repeats (*Johnson and Jasin, 2000*). We tested whether farrerol potentially affects SSA, thereby causing a loss of genome integrity. We employed two previously reported reporters HRF, which measures the efficiency of gene conversion, crossover and SSA (*Figure 1—figure supplement 3A*), and HRIF, which measures only gene conversion and crossover (*Figure 1—figure supplement 3B*). Therefore, SSA efficiency can be quantified with HRF minus HRIF (*Mao et al., 2012*). The two reporters were integrated into HCA2-hTERT cells, and a pool of colonies containing chromosomally integrated reporters were mixed for further analysis. Consistent with a previous report that RAD51 is not involved in SSA pathway (*Benitez et al., 2018*; *Bennardo et al., 2008*; *Mendez-Dorantes et al., 2018*), we found that activating RAD51 with RS-1 had no effect on SSA efficiency. Surprisingly, treating cells with farrerol significantly suppressed SSA efficiency by 3.3-fold and 1.9-fold at concentrations of 5 µM and 10 µM respectively (*Figure 1—figure supplement 3C*), suggesting that farrerol might help maintain genome integrity by avoiding the aberrant recombination between the abundant repetitive sequences across genomes.

## Farrerol promotes the efficiency of SpCRISPR/Cas9-mediated gene targeting in HEK 293FT cells

Precisely inserting an exogenous gene into a specific locus requires efficient and faithful HR directed repair (*Pinder et al., 2015*). We therefore set out to examine whether farrerol enhances the efficiency of SpCRISPR/Cas9-mediated gene targeting. HEK 293FT cells were pretreated with farrerol or two positive control compounds SCR7 (*Chu et al., 2015*; *Li et al., 2017a*) and RS-1 (*Riesenberg and Maricic, 2018*) at the indicated concentrations for 24 hr, followed by co-transfection with vectors encoding Cas9 and sgRNA targeting the *AAVS1* site, and a donor plasmid

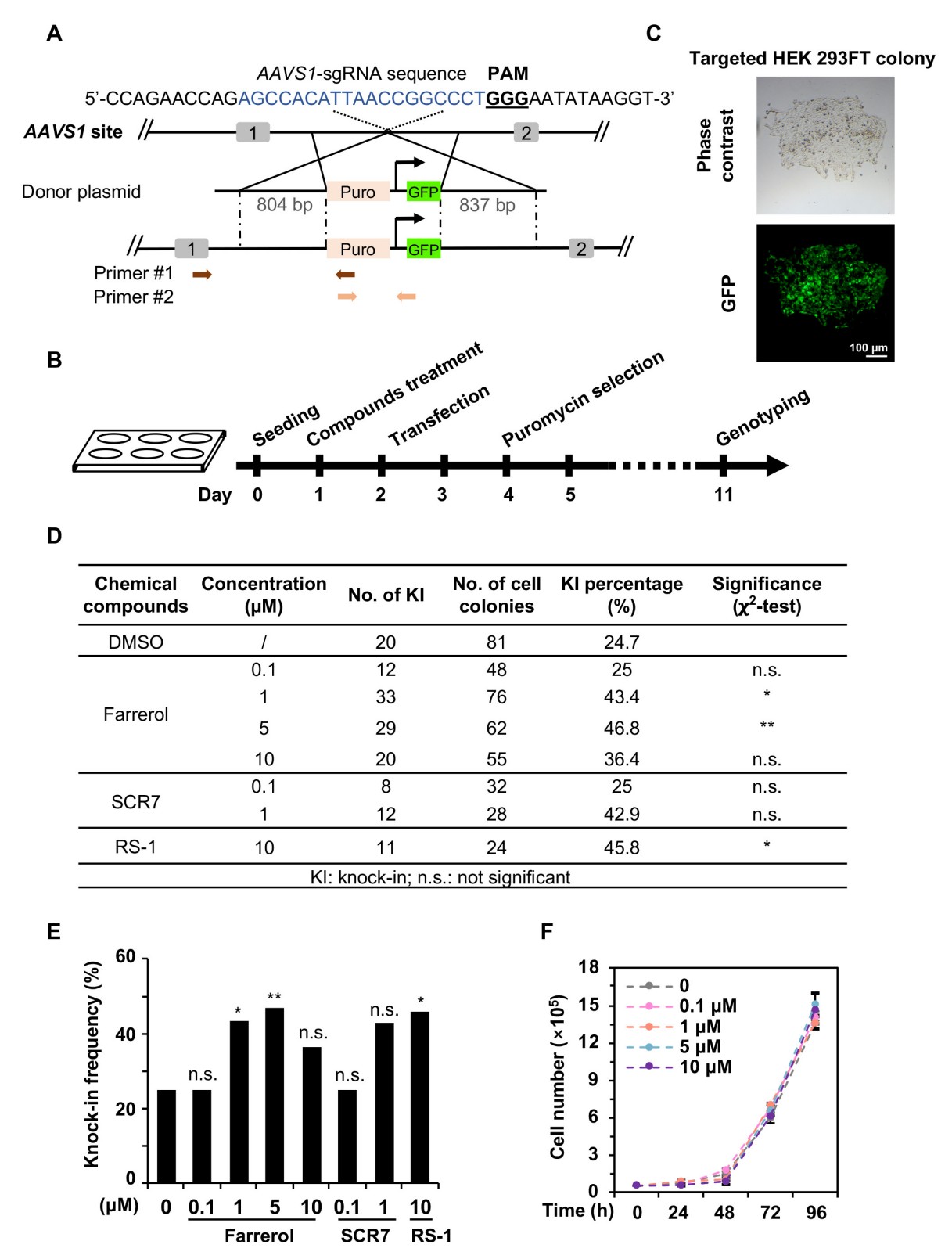

**Figure 2.** Farrerol promotes SpCRISPR/Cas9-mediated knock-in efficiency in human cells. (A) Schematic diagram of the gene targeting strategy at the human *AAVS1* locus. A donor vector containing a promoter-less p2A-puromycin and GFP gene driven by the CAG promoter was designed for targeting the *AAVS1* locus. The underlined trinucleotide represents the PAM, and the sgRNA targeting site is labeled in blue. Puro stands for puromycin. Two pairs of primers used for genotyping are indicated by arrows. The length of the left and right homologous arm is 804 bp and 837 bp,

*Figure 2 continued*

respectively. Primer sequences are listed in the *Table 1*. (B) Workflow of gene targeting at the *AAVS1* locus in HEK 293FT cells. (C) Representative microscopy images of successfully targeted GFP⁺ HEK 293FT cells post 1 μg/mL puromycin selection. (D–E) Effect of different small molecules on gene knock-in frequency at the human *AAVS1* locus. The surviving colonies post puromycin selection were further validated by genotyping using PCR. (F) Growth curve of HEK 293FT cells treated with the indicated doses of farrerol. For (D) and (E), $\chi^2$-test was used for statistical analysis. Error bars in (F) represent the s.d. and t-test was used for statistical analysis. *p<0.05, **p<0.01, n.s., not significant. All experiments were repeated at least three times. The online version of this article includes the following source data and figure supplement(s) for figure 2:

**Source data 1.** Summary of knock-in efficiency in HEK 293FT cells.
**Figure supplement 1.** The effects of small molecule treatment on knock-in efficiency at the *AAVS1* locus in HEK 293FT cells.

containing a promoter-less puromycin and a GFP gene driven by the CAG promoter (*Figure 2A*). At 48 hr post transfection, cells were supplemented with 1 μg/mL puromycin, and puromycin-resistant and GFP⁺ colonies were then quantified (*Figure 2B–C*). We further analyzed the knock-in efficiency using PCR with primers as indicated (*Table 1*; *Figure 2A*). We found that farrerol significantly stimulated the knock-in efficiency by ~2.0 fold at the concentration of 1 and 5 μM, which was similar to those of RS-1 or SCR7 treatment (*Figure 2D–E*). In addition, we did not observe any significant effect on cell proliferation in the presence of farrerol at the indicated concentrations (*Figure 2F*), suggesting a low toxicity of this compound.

**Table 1.** PCR primer sequences.

| Target name | Primer name | Sequence (5'>3') | Description | Reference |
|---|---|---|---|---|
| *AAVS1* | HR-AAVS1-F | gccgtctctctcctgagt | Primer 1# | PMID:23287722 |
| *AAVS1* | HR-Puro-R | gtgggcttgtactcggtcat | | |
| *AAVS1* | GFP-F | acgtaaacggccacaagttc | Primer 2# | This paper |
| *AAVS1* | GFP-R | gaactccagcaggaccatgt | | |
| *Actb* | mActb-HR-F | ccatctacgagggctatgct | 5' junction | This paper |
| *Actb* | mActb-HR-R | gtgggcttgtactcggtcat | | |
| *Actb* | 3' Actb-puro-F | gtgtctctcactcggaaggac | 3' junction | |
| *Actb* | 3'inner-R | gcctaggtttctggaggagt | | |
| *Actb* | 5'outer-F | ccctgagtgtttcttgtggc | 5' junction | PMID:28524166 |
| *Actb* | 5'outer-R | tggagccgtacatgaactga | | |
| *Actb* | 5'inner-F | ccatctacgagggctatgct | | |
| *Actb* | 5'inner-R | tgaagcgcatgaactccttg | | |
| *Actb* | 3'outer-F | gccccgtaatgcagaagaag | 3' junction | |
| *Actb* | 3'outer-R | aggtagtgttagtgcaggcc | | |
| *Actb* | 3'inner-F | ctacgacgctgaggtcaaga | | |
| *Actb* | 3'inner-R | gcctaggtttctggaggagt | | |
| *Cdx2* | 5'outer-F | acttggacagagaaagagcgatt | 5' junction | PMID:28524166 |
| *Cdx2* | 5'outer-R | tccatgtgcaccttgaagc | | |
| *Cdx2* | 5'inner-F | aacaaaggtccagtctacgcat | | |
| *Cdx2* | 5'inner-R | ggccatgttatcctcctcgc | | |
| *Cdx2* | 3'outer-F | gacggccccgtaatgcagaa | 3' junction | |
| *Cdx2* | 3'outer-R | tagcttgcaaccagagaagatgt | | |
| *Cdx2* | 3'inner-F | ctacgacgctgaggtcaaga | | |
| *Cdx2* | 3'inner-R | cgacttcccttcaccatacaac | | |

The online version of this article includes the following source data for Table 1:
**Source data 1.** PCR primer sequences.

More strictly, we further examined whether farrerol could improve the efficiency of SpCRISPR/Cas9-mediated gene targeting using a different targeting vector. The p2A-mCherry-WPRE-polyA vector was then generated which also targeted the *AAVS1* locus (*Figure 2—figure supplement 1A*). Knock-in efficiency was measured by scoring the number of mCherry⁺ cells with FACS (*Figure 2—figure supplement 1B*). Consistently, we found that farrerol could significantly improve the targeting efficiency in this system at concentrations ranging from 0.1 μM to 10 μM (*Figure 2—figure supplement 1B–C*).

Taken together, these data reveal that farrerol improves knock-in efficiency in HEK 293FT cells.

## Farrerol promotes SpCRISPR/Cas9-mediated gene knock-in in mouse ESCs

Because efficient gene targeting in ESCs holds great potential for further application in both basic research and clinical medicine, we set out to test whether farrerol could enhance the efficiency of SpCRISPR/Cas9-mediated insertions in mouse ESCs. We first examined the insertion efficiency at the *Actin β* (*Actb*) locus, using an sgRNA as previously described (*Yao et al., 2017*). Mouse ESCs were pretreated with farrerol, SCR7 or RS-1 for 24 hr, followed by a co-transfection with vectors encoding Cas9, sgRNA targeting the *Actb* site, and a donor plasmid containing a promoter-less p2A-puromycin and GFP gene driven by the CAG promoter (*Figure 3A*). Successful insertion mediated by HR directed repair in this system results in an *Actb*-puromycin-CAG-GFP fusion protein, converting cells to puromycin resistant and GFP positive. Then, these puromycin-resistant colonies could be stained with Coomassie blue and counted to score insertion efficiency (*Figure 3B*). Meanwhile, GFP⁺ colonies could be observed under the fluorescence microscope (*Figure 3C*). Strikingly, we found that treating cells with an increasing concentration of farrerol significantly promoted knock-in efficiency by as high as 2.8-fold (10 μM) while both SCR7 and RS-1 had only mild effects on insertion efficiency (*Figure 3D*). Furthermore, we picked individual mouse ESC colonies, and confirmed the correct insertion into the *Actb* locus by PCR and Sanger sequencing (*Figure 3E*). Together, these data demonstrate that farrerol dramatically stimulates the efficiency of SpCRISPR/Cas9-mediated gene targeting into *Actb* locus of mouse ESCs.

In addition to the *Actb* locus, we further tested if pretreatment of farrerol facilitated the SpCRISPR/Cas9-mediated targeting into two other loci, *Rosa26* and *Sox2*. A p2A-BFP-WPRE donor was designed to target the *Rosa26* locus in mouse ESCs (*Figure 3—figure supplement 1A*). On day three post transfection of plasmids containing the donor and Cas9 along with sgRNA targeting the *Rosa26* site, the percentage of BFP⁺ cells was measured by FACS. Consistently, treating cells with farrerol significantly improved knock-in efficiency, with the largest increase of 2.9-fold observed at the concentration of 5 μM (*Figure 3—figure supplement 1B–C*). Additionally, a 2.0-fold improvement in knock-in efficiency at the *Sox2* site could be observed after farrerol treatment at the concentration of 5 μM, by examining with a previously reported p2A-mCherry donor (*Yao et al., 2017*; *Figure 3—figure supplement 2*). Collectively, farrerol can efficiently promote HR at multiple loci in mouse ESCs.

## Farrerol has no obvious adverse effects on genomic stability in human fibroblasts or mouse ESCs

To investigate whether treating cells with farrerol would lead to a destabilized genome, we first performed immunostaining experiments with antibodies against γH2AX and 53BP1 to examine the potential genotoxic side effects of each small molecule. We observed a mild decrease in γH2AX foci number or fluorescence intensity, and no obvious change in 53BP1 foci number in farrerol-treated HCA2-hTERT fibroblasts or mouse ESCs (*Figure 4A–D*; *Figure 4—figure supplement 1*). However, treating cells with SCR7 and RS-1 resulted in the increase of the percentage of γH2AX positive cells, suggesting the genotoxic side effects of these two molecules on cells (*Figure 4—figure supplement 1A–B*). Comet assay was also performed to examine if farrerol had any negative effect on genome integrity, and no significant change was noticed in the presence of farrerol (*Figure 4E*). By contrast, although fairly mild, we observed a significant increase in genomic instability in mouse ESCs treated with SCR7 (*Figure 4E*), which inhibits the NHEJ pathway. To further explore the potential influence of drug treatment on large-scale chromosomal rearrangements, we further performed karyotyping analysis. The result showed that farrerol treatment had no obvious influence on the karyotype of

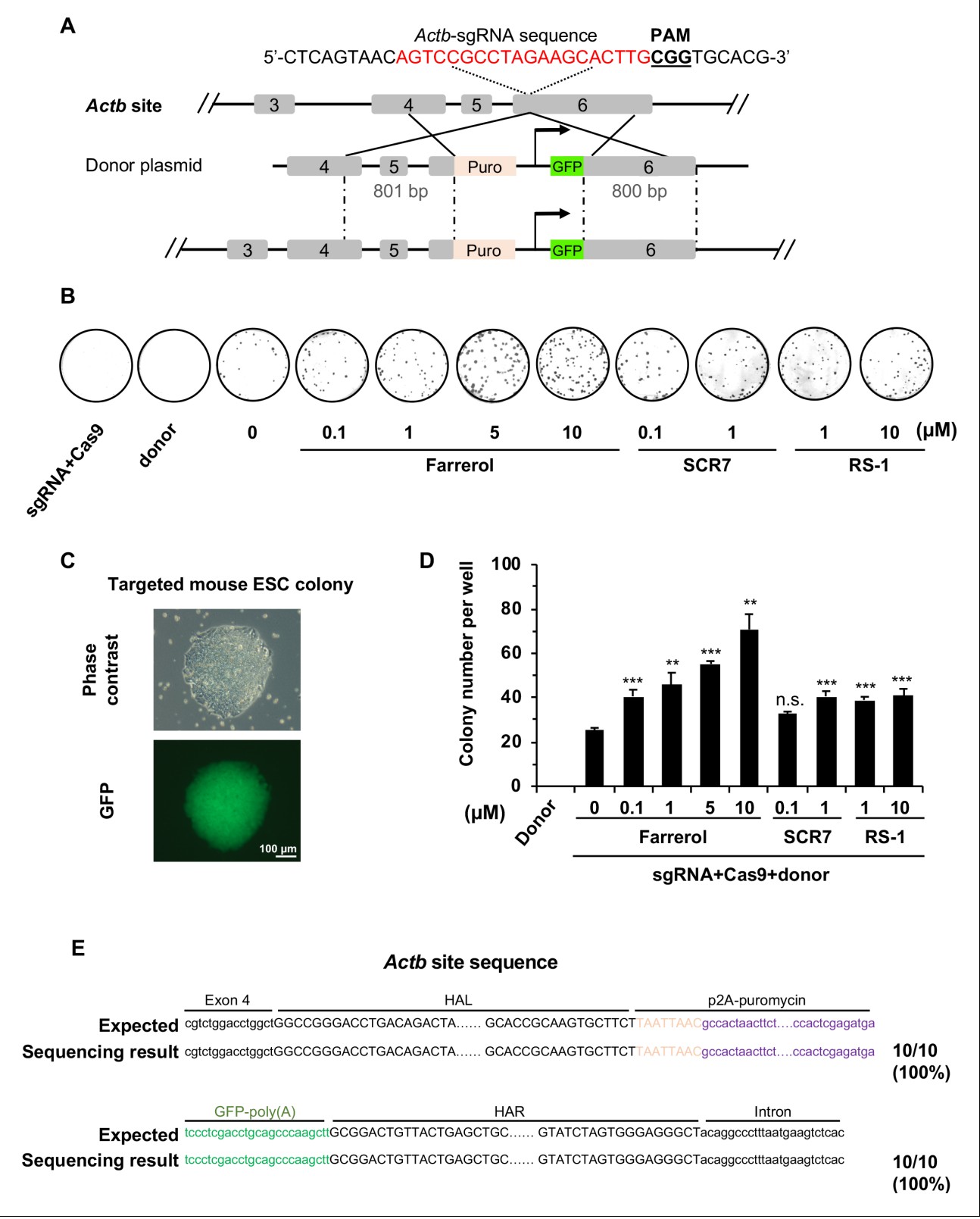

**Figure 3.** Farrerol promotes SpCRISPR/Cas9 mediated gene targeting efficiency in mouse ESCs. (**A**) Schematic diagram of the gene targeting strategy at the mouse *Actb* locus. A donor vector containing a promoter-less p2A-puromycin (Puro) and GFP gene driven by the CAG promoter was designed for targeting the mouse *Actb* locus. The underlined trinucleotide represents the PAM, and the sgRNA targeting site is labeled in red. The length of the left and right homologous arm is 801 bp and 800 bp, respectively. (**B**) Representative images of Coomassie blue stained puromycin resistant E14 cells

*Figure 3 continued on next page*

*Figure 3 continued*

which were successfully knocked in. Transfection of donor only or the mixture of sgRNA and Cas9 were set as the negative control. (C) Representative microscopy images of successfully targeted E14 cells with GFP expression. (D) Effect of different small molecules on gene targeting frequency at the *Actb* locus in mouse ESCs. The knock-in efficiency was measured by counting the cell colonies which were resistant to puromycin. (E) The Sanger sequencing results of the 5' and 3' junction regions of successfully knocked-in cells treated with farrerol. HAL stands for the left homologous arm and HAR stands for the right homologous arm. Error bars represent the s.e.m. **p<0.01, ***p<0.001, n.s., not significant, t-test. All experiments were repeated at least three times.

The online version of this article includes the following source data and figure supplement(s) for figure 3:

**Source data 1.** Summary of knock-in efficiency in mouse ESCs.
**Figure supplement 1.** The effects of small molecule treatment on knock-in efficiency at the *Rosa26* locus in mouse ESCs.
**Figure supplement 2.** The effects of small molecule treatment on knock-in efficiency at the *Sox2* locus in mouse ESCs.

mouse ESCs after gene editing. And more surprisingly, treating mouse ESCs with SCR7 or RS-1 for only 48 hr caused a significant decrease in the percentage of cells with normal karyotypes (*Figure 4F*). Furthermore, we also performed EdU incorporation experiments to examine if farrerol affected DNA replication in mouse ESCs. In agreement with the analysis in HEK 293FT cells (*Figure 2F*), we did not observe any significant change in DNA replication in mouse ESCs (*Figure 4—figure supplement 2*).

In summary, these results indicate that farrerol is a safe compound when used to improve the efficiency of precise genome integration.

## Farrerol effectively promotes targeted integration and supports the efficient generation of gene-targeting mice with germline transmission

To further test the utility of farrerol in the embryo system and whether it could facilitate the generation of gene targeted animals, we first examined its potential toxicity. Mouse zygotes were cultured in media containing either farrerol, RS-1 or SCR7, and we then examined whether these compounds had any impact on the *in vitro* development potential of mouse embryos. We found that farrerol could fully support the mouse embryos through the hatching of blastocysts as did control and RS-1; by contrast, SCR7 treatment significantly impaired the blastocyst production rate (*Figure 5A–B*; *Figure 5—figure supplement 1A*). Considering the SpCRISPR/Cas9-mediated cleavage of DNA and the subsequent HR directed repair occurs and is primarily completed during the 1 cell to 2 cell stage, we then cultured the mouse zygotes in the indicated small molecule containing media through the late 2 cell stage. The results indicated that short-period treatment with the three compounds had nearly no effect on the early embryo development and more than 95% of the 2 cell embryos could develop into the blastocyst stage (*Figure 5—figure supplement 1B*). However, during the zygotes stage where DSBs were induced due to the injection of Cas9 mRNA and sgRNA targeting the *Cdx2* locus, treatment with the three compounds yielded different results. Only 62.5% of SCR7 treated embryos could develop into the blastocyst stage, probably due to the inhibition of NHEJ. By contrast, farrerol treated embryos showed an 82.4% blastocyst production rate (*Figure 5C*; *Figure 5—figure supplement 1C*).

To further investigate whether farrerol could improve knock-in efficiency in mouse embryos, we injected Cas9 mRNA, sgRNA targeting the *Actb* gene and p2A-mCherry donor cassette into mouse zygotes (*Figure 5D–E*; *Yao et al., 2017*). Then the injected zygotes were treated with farrerol at a concentration of 0 (control), 0.05 µM, 0.1 µM, or SCR7 (20 µM), RS-1 (7.5 µM) as previously reported (*Song et al., 2016*). At late 2 cell stage, these embryos were picked out and cultured in normal media until the blastocyst stage, and the integration efficiencies were assessed by mCherry$^+$ fluorescence at E4.5 (*Figure 5F*). Interestingly, a much higher rate of mCherry$^+$ blastocysts was observed in the presence of farrerol (0.05 µM, 27.3%; 0.1 µM, 21.0%), than in the control treatment (14.0%) (*Figure 5F–G*). Importantly, this efficiency is comparable to or even higher than that mediated by HMEJ (Homology-mediated end joining, 22.7%) and MMEJ (Microhomology-mediated end joining, 11.9%) donors (*Yao et al., 2017*). By contrast, the knock-in efficiency was only slightly improved after SCR7 (19.2%) or RS-1 (16.5%) treatment, which is in agreement with a recent study (*Song et al., 2016*). We then examined the integration fidelity by genotyping the integration sites of individual blastocysts. Consistently, PCR amplification of both the 5' and 3' junctions revealed that farrerol treatment led to a higher in-frame integration rate (farrerol: 0.05 µM, 27.0%;

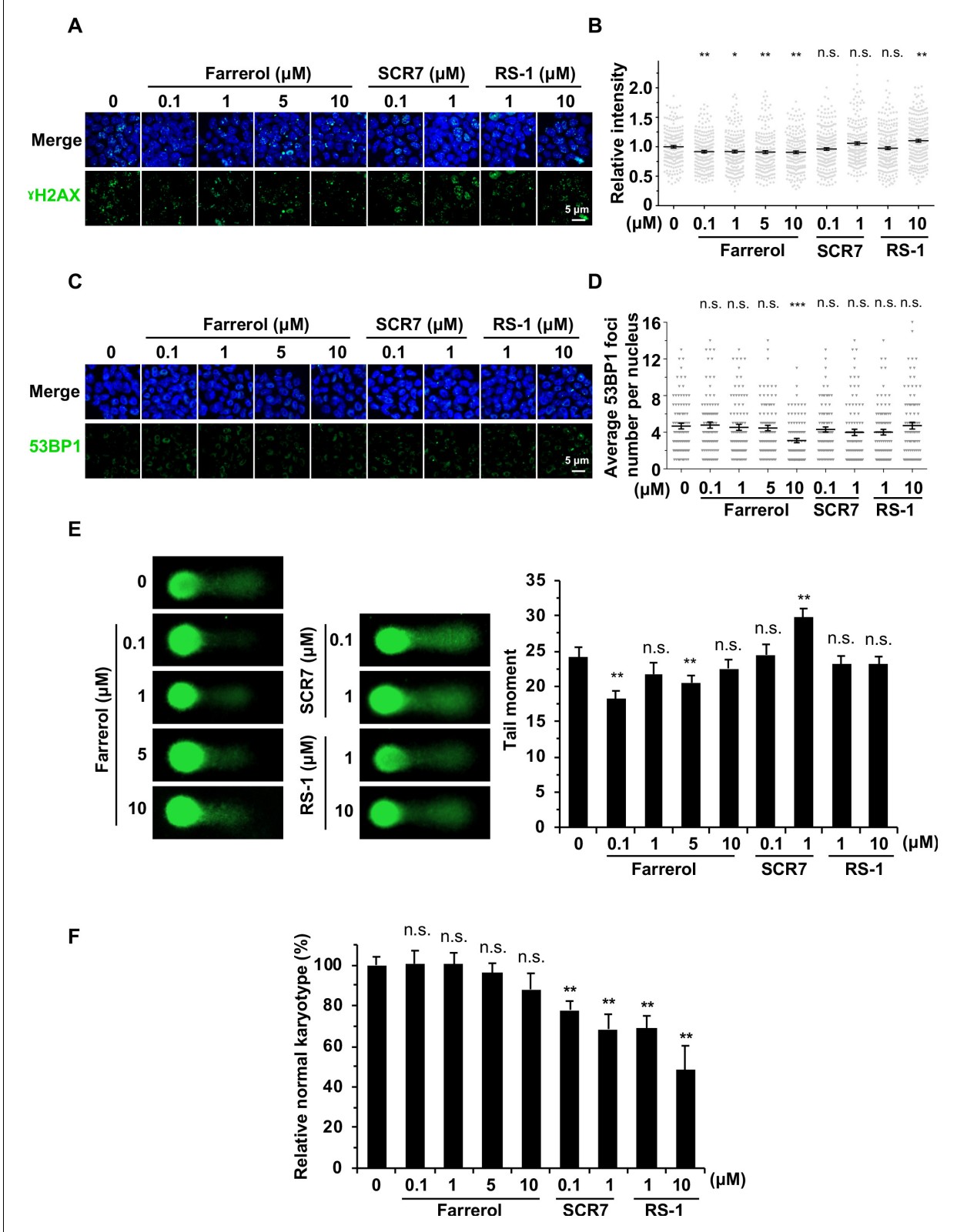

**Figure 4.** The effects of small molecule treatment on genomic stability in mouse ESCs. (**A**) Representative images of γH2AX immunostaining in mouse ESC E14 treated with indicated doses of small molecules. (**B**) Analysis of relative intensity of γH2AX in mouse ESCs in (**A**). The results were normalized to those in control group. Each dot represents the relative fluorescence intensity of a single nucleus. (n ≥ 180 single nucleus). (**C**) Representative images of 53BP1 immunostaining in mouse ESC E14 treated with indicated doses of small molecules. (**D**) Quantification of 53BP1 foci numbers in (**C**). At least

*Figure 4 continued on next page*

*Figure 4 continued*

50 cells were included for each group. (**E**) Representative images of alkaline comet assay of mouse ESC E14 treated with indicated doses of molecules for 24 hr (left panel). The tail moments of at least 50 cells for each group were quantified using Cometscore software (Sumerduck, VA, USA) (right panel). (**F**) Karyotyping analysis indicated that 48 hr farrerol treatment post spCRSIPR/Cas9 mediated editing did not greatly affect karyotypes (40, XY) of the Oct4-ΔPE-GFP transgenic C57BL/6 × PWK ESCs. Error bars in (**B**), (**D**) and (**E**) represent the s.e.m. Error bars in (**F**) represent the s.d. *p<0.05, **p<0.01, ***p<0.001, n.s., not significant, t-test.

The online version of this article includes the following source data and figure supplement(s) for figure 4:

**Source data 1.** The genotoxicity of small compounds treatment.
**Figure supplement 1.** The effects of small molecule treatment on genomic stability in HCA2-hTERT cells.
**Figure supplement 2.** The effects of small molecule treatment on cell proliferation.

0.1 µM, 23.0%) in comparison to the control group (13.0%). However, treatment with SCR7 (15.1%) or RS-1 (17.8%) were shown to have limited effect (*Figure 5H*; *Figure 5—figure supplement 1D*).

To rule out the possibility that the promotion of the SpCRISPR/Cas9-mediated genome targeting efficiency is position-dependent in mouse embryos, we further examined the knock-in efficiency at the *Cdx2* locus, by fusing a p2A-mCherry cassette to the C-terminus of the trophectoderm (TE) marker gene (*Figure 6A*). Successful integration into *Cdx2* locus results in mCherry positivity, which can be observed in trophoblast cells in the blastocyst. Embryos treated by farrerol showed a repeatable and remarkable increase of mCherry$^+$ fluorescence signals in TE by 57.4% (0.05 µM) and 68.3% (0.1 µM) in comparison to that in control group. As a comparison, SCR7 and RS-1 improved the targeting efficiency by 54.3% and 37.4% respectively (*Figure 6B–C*). Genotyping analysis further confirmed the capacity of farrerol to robustly stimulate SpCRISPR/Cas9-mediated gene editing (*Figure 6D*; *Figure 6—figure supplement 1A*).

In order to test if farrerol potentially affected the late stages of embryonic development and the generation of knock-in mice, we transplanted farrerol-treated microinjected 2 cell stage embryos into pseudo-pregnant mice (*Figure 6—figure supplement 1B*). In the farrerol treated group, 12 out of 33 mice showed an in-frame integration (36.4%; potential founders #1, #2, #3, #7, #9, #16, #18, #21, #23, #26, #27, #32), which was confirmed by the PCR amplification of both the 5' and 3' junctions at *Cdx2* locus (*Figure 6E*; *Figure 6—figure supplement 1C*). Sanger sequencing data further supported this conclusion (*Figure 6—figure supplement 1D*). By contrast, only one knock-in mouse was identified in a total 53 mice generated in control group. In addition, as a further comparison, no knock-in mice were detected by using the HMEJ donor (*Yao et al., 2017*; *Figure 6E*). Moreover, *Cdx2*-mCherry mice generated from farrerol treated embryos had germline transmission abilities and the allele with the exogenous gene knocked in could be stably transmitted to the progeny generation (F1) with Mendel's law of segregation, after being crossed with wild-type C57BL/6 mice (*Figure 6E*). Furthermore, homozygous *Cdx2*-mCherry mice could be successfully produced and maintained. And all the blastocysts from these mice after crossing showed the strong mCherry$^+$ fluorescence signals in the trophectoderm (*Figure 6F*). At the E7.5 stage, these mCherry signals were restricted to extraembryonic ectoderm (ExE) (*Figure 6G*). At the stage of E9.5, *Cdx2* mCherry signals could be observed in embryonic tissues, principally in the posterior part of the gut, tail bud and the caudal part of the neural tube as previously reported (*Figure 6G*; *Beck et al., 1995*).

Since the glycoprotein matrix, termed the zona pellucida (ZP), surrounding mammalian eggs physically separates the embryo from the external environment which may restrict the actual working concentration of this small molecule, we then asked whether the integration efficiency could be further improved by direct injection of farrerol into the cytoplasm of the zygote. Thus, small molecules (farrerol at final concentration of 0.05 µM, 0.1 µM; SCR7 at final concentration of 1 µM; RS-1 at final concentration of 7.5 µM) were individually added to the CRISPR/Cas9 mixture (Cas9 mRNA, sgRNA and donor) for microinjection into the zygotes and the injected embryos were cultured in normal medium into blastocysts. Intriguingly, in comparison to control group, we observed a significantly increased rate of gene editing at the *Actb* site (0.05 µM, by 2.3-fold; 0.1 µM, by 2.6-fold) and *Cdx2* locus (0.05 µM, by 2.0-fold; 0.1 µM, by 2.3-fold) after co-injection of farrerol, while co-injecting the other two compounds had mild effects on integration efficiency (*Figure 6—figure supplement 2A–B*). The genotyping analysis further confirmed that farrerol could promote gene editing at these two genomic loci (*Figure 6—figure supplement 2C–D*).

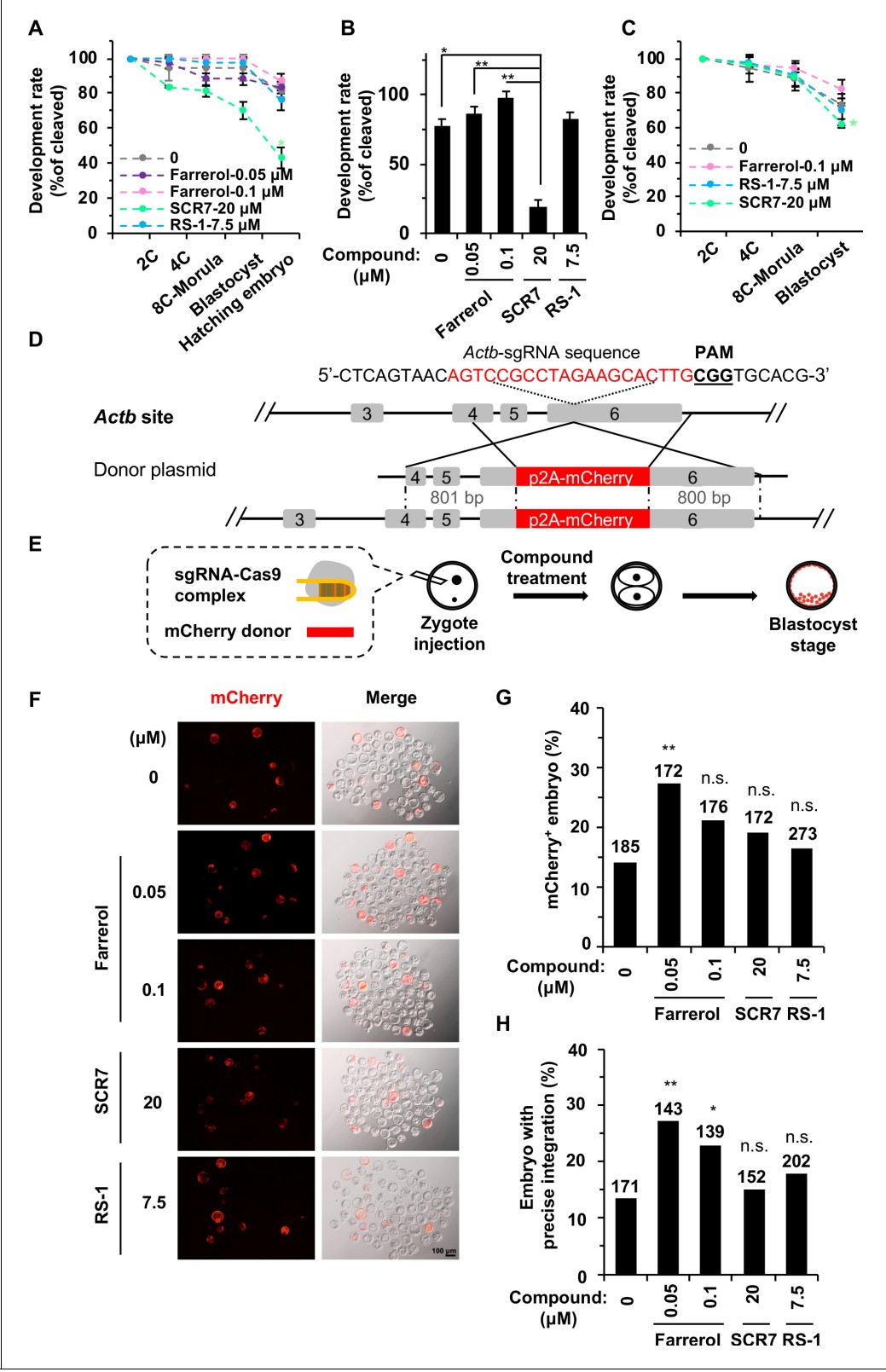

**Figure 5.** Farrerol promotes gene targeting in mouse embryos. (**A**) *In vitro* development potential of mouse embryos treated with indicated doses of compounds during the whole embryonic stages. (**B**) The development rate of hatching blastocysts at E4.5 treated with indicated doses of compounds. The data is related to (**A**). (**C**) Effect of compounds treatment for 24 hr on *in vitro* development potential of mouse embryos upon DSBs were induced by Cas9 mRNA and sgRNA targeting the *Cdx2* locus. (**D**) Schematic diagram of the gene targeting strategy at the *Actb* locus in mouse embryos. A
*Figure 5 continued on next page*

Figure 5 continued

donor vector containing a promoter-less p2A-mCherry was designed for targeting the *Actb* locus. The underlined trinucleotide represents the PAM, and the sgRNA targeting site is labeled in red. The length of the left and right homologous arm is 801 bp and 800 bp, respectively. (**E**) Diagram of the methods for gene targeting efficiency analysis in mouse blastocysts. (**F**) Representative fluorescence images of gene-edited mouse embryos at the *Actb* locus at the blastocyst stage. (**G–H**) Effect of indicated small compounds on gene knock-in frequencies at the *Actb* locus. Knock-in frequency was indicated by the percentage of mCherry$^+$ blastocysts in (**G**), and was confirmed by PCR genotyping analysis using primers amplifying the flanks of the *Actb* site in (**H**). Number above each bar, total blastocysts analyzed. Error bars in (**A**), (**B**) and (**C**) represent the s.d. and t-test was used for statistical analysis. For (**G**) and (**H**), $\chi^2$-test was used for statistical analysis. *p<0.05, **p<0.01, n.s., not significant.

The online version of this article includes the following source data and figure supplement(s) for figure 5:

**Source data 1.** Summary of knock-in efficiency in blastocysts.
**Figure supplement 1.** The effects of small molecule treatment on embryo development.

Cumulatively, these results demonstrate that farrerol can enhance in-frame integration of exogenous donor DNA and efficiently generate knock-in mice with germline transmission capacity.

## Discussion

Recently developed exogenous enzyme-based genome editing tools have greatly improved the efficiency of gene knock-out as the generated DSBs can be repaired by the error-prone and cell cycle-independent NHEJ pathway, resulting in non-functional genes (*Guo et al., 2018*; *Maeder and Gersbach, 2016*). In contrast, precisely knocking genetic material into an endogenous locus requires an active HR pathway, which is not a predominant DSB repair pathway in human cells (*Mao et al., 2008a*; *Mao et al., 2008b*). Therefore, promoting HR directed repair or suppressing the NHEJ pathway have been explored as tools for improving the efficiency of SpCRISPR/Cas9-mediated gene targeting (*Smirnikhina et al., 2019*). Quite different from other screening platforms applied for searching for gene targeting activators, here, we created the screening platform based on DSB repair outcomes. The advantages of this system are that (1) it does not require any transfection of exogenous genes, and therefore avoids any potential interference resulting from transfections, such as changes in cell cycle stages, in which DSB repair pathways differ; (2) it can be easily scored either on FACS or on a high throughput automated high-content IN Cell imaging system; (3) the screened compounds affecting the efficiency of the two DSB repair pathways can be applied to other purposes including targeting cancer alone or in combination with other therapeutic methods, and delaying the onset of aging by improving genome integrity.

Our data clearly indicate that farrerol promotes HR through facilitating the recruitment of RAD51, the critical recombinase involved in HR, to DSB sites, but has no obvious effect on NHEJ. This result explains our observation that farrerol promotes SpCRISPR/Cas9-mediated targeting efficiency at different genomic loci in different cell types, indicating that farrerol holds great potential in applications related to SpCRISPR/Cas9-mediated genome editing. Among a number of compounds that stimulate the efficiency of gene targeting, RS-1 and SCR7 are the best-characterized and the most commonly used. RS-1 physically interacts with RAD51 to promote HR (*Jayathilaka et al., 2008*). Surprisingly, the promotion of gene targeting by RS-1 is dependent on the size of the homologous arm of the targeting vectors (*Pan et al., 2016*; *Pinder et al., 2015*), and the type of cells (*Song et al., 2016*; *Zhang et al., 2017*). It is hypothesized that the promotion of gene targeting by SCR7 is through suppressing the canonical NHEJ (c-NHEJ) pathway by blocking the enzymatic activity of DNA LIG4, forcing cells to choose the other DSB repair pathway – HR (*Greco et al., 2016*; *Srivastava et al., 2012*). Although some of our results confirmed previous findings that supplementing SCR7 improves the efficiency of gene targeting (*Maruyama et al., 2015*; *Singh et al., 2015*; *Song et al., 2016*), the effects are rather mild and it does not always function at the given genomic locus or in different types of cells. This can be explained by the fact that in addition to c-NHEJ and HR, alternative NHEJ (alt-NHEJ) can also repair DSBs. Treating cells with SCR7 actually leaves DSBs repaired either by HR and alt-NHEJ. Whether an optimized treatment with farrerol – such as concentration, treatment time or combinations of these compounds can further improve gene targeting efficiency remains to be further determined. Moreover, a comprehensive and thorough study on the mechanism by which farrerol promotes HR would also help the future application of this compound in genome editing.

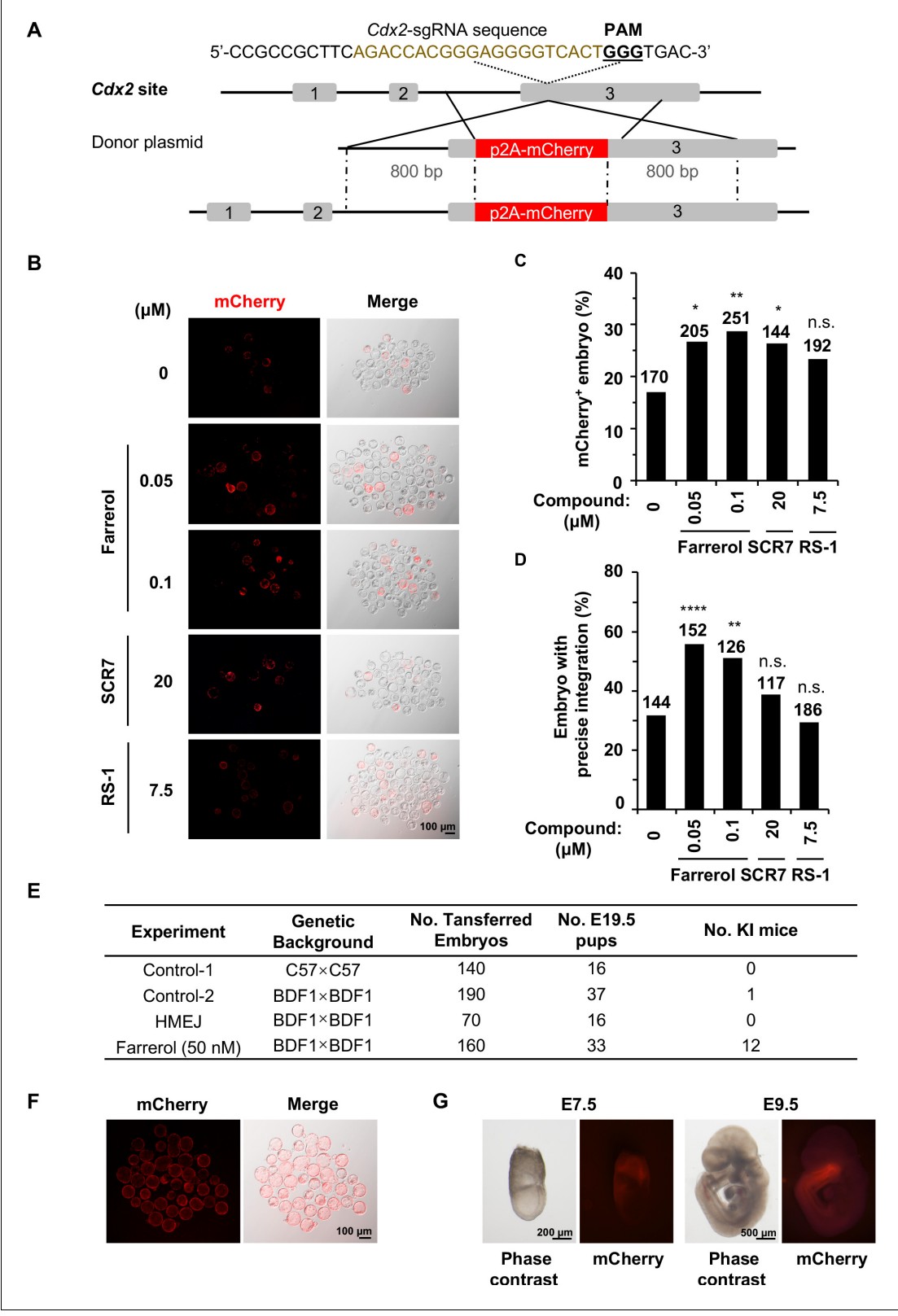

**Figure 6.** Farrerol promotes the generation of gene-targeted mice with germline transmission. (**A**) Schematic diagram of the gene targeting strategy at the *Cdx2* locus in mouse embryos. A donor vector containing a promoter-less p2A-mCherry was designed for targeting the *Cdx2* locus. The underlined trinucleotide represents the PAM, and the sgRNA targeting site is labeled in brown. The length of both the left and right homologous arm are 800 bp. (**B**) Representative fluorescence images of gene-edited mouse embryos at the *Cdx2* locus at the blastocyst stage. (**C–D**) Effect of different small

*Figure 6 continued on next page*

*Figure 6 continued*

molecules treatment on gene knock-in frequency at the *Cdx2* locus. Knock-in frequency was indicated by the percentage of mCherry$^+$ blastocysts in (**C**), and was confirmed by PCR genotyping analysis using primers amplifying the flanks of the *Cdx2* site in (**D**). Number above each bar, total blastocysts analyzed. (**E**) Effect of different knock-in strategies on generation of gene-targeting mice. The microinjected 2 cell stage embryos with or without farrerol treatment was transplanted into the pseudo-pregnant mice. The HMEJ mediated knock-in assay was applied as a control. The founder mice were genotyped for gene-targeting frequency analysis. The germline transmission abilities of founder mice were also validated. (**F**) Representative fluorescence images of blastocysts from homozygous *Cdx2*-mCherry mice. (**G**) Representative fluorescence images of *Cdx2* mCherry signals in embryos at the stage of E7.5 and E9.5 after homozygous *Cdx2*-mCherry mice mating with wild-type mice. *p<0.05, **p<0.01, ****p<0.0001, n.s., not significant, $\chi^2$-test.

The online version of this article includes the following source data and figure supplement(s) for figure 6:

**Source data 1.** Generation of gene-targeted mice with germline transmission.
**Figure supplement 1.** Farrerol promotes precise integration in blastocysts and founder mice.
**Figure supplement 2.** The effects of small molecule intra-cytoplasmic injection on knock-in efficiency.

Since HR and NHEJ are two competitive pathways when repairing DSBs particularly in the S/G2 phase with sister chromatids available (*Mao et al., 2008b*), why does not the activation of HR pathway by farrerol cause the decrease in NHEJ? Upon DSB induction, several consequences might occur: (1) DSBs are not repaired and cells undergo cellular senescence or apoptosis; (2) DSBs are repaired by canonical NHEJ or alternative NHEJ; (3) DSBs are repaired by three sub-pathways of HR, gene conversion, crossover and SSA. Since we also observed that farrerol suppresses the mutagenic SSA pathway, it is possible that without affecting the two sub-NHEJ pathways, farrerol promotes the two precise HR pathways – gene conversion and crossover – by suppressing SSA pathway. Another possibility is that farrerol might tip cells with DSBs away from entering senescence or apoptosis and towards DSB repair by HR.

Safety is one of the most noteworthy issues in the field of genome editing. Our data suggest that farrerol exhibits no detectable adverse effect on the normal development of mouse blastocysts *in vitro*. More importantly, healthy founder mice with the precise genomic edit of interest could be efficiently generated, and the edited allele could be successfully transmitted to their offspring. Although some literature has reported that high concentration of SCR7 treatment increased the HR efficiency by as much as 10-fold in mouse embryos (*Singh et al., 2015*) and mammalian cells (*Li et al., 2017a*), our data indicated that SCR7 at such a high concentration would lead to adverse effects on karyotype and blastocyst development. The reason might lie in that inhibiting c-NHEJ might lead to inefficient and late repair of DSBs (*Chen et al., 2008*; *Vartak and Raghavan, 2015*), resulting in abnormal development. In addition, blocking c-NHEJ may destabilize genomes, increasing the incidence of tumorigenesis and the onset of premature aging (*Chen et al., 2008*; *Sekiguchi et al., 1999*; *Vogel et al., 1999*). Moreover, the further application of compounds activating HR directed repair should also be cognizant of the possibility that aberrantly activated HR might be beneficial to tumor cells, and may disrupt the genome integrity by aberrantly recombining the abundantly available repetitive sequences across the genomes in normal cells (*Putnam et al., 2009*; *Sasaki et al., 2010*). For instance, increased expression of RAD51 destabilizes genomes by stimulating aneuploidy and making chromosomal rearrangements more likely (*Richardson et al., 2004*). Future studies on whether and how a short-term treatment of RS-1 or farrerol affects genome integrity are still warranted.

## Materials and methods

### Key resources table

| Reagent type (species) or resource | Designation | Source or reference | Identifiers | Additional information |
|---|---|---|---|---|
| Genetic reagent (*M. musculus*) | C57BL/6n | Beijing Vital River Laboratory | Stock No.: 213 | |

*Continued on next page*

*Continued*

| Reagent type (species) or resource | Designation | Source or reference | Identifiers | Additional information |
|---|---|---|---|---|
| Genetic reagent (*M. musculus*) | ICR | Beijing Vital River Laboratory | Stock No.: 201 | |
| Genetic reagent (*M. musculus*) | BDF1 | This paper | / | The BDF1 hybrid mice were obtained from mating female C57BL/6n mice with male DBA/2 mice. |
| Genetic reagent (*M. musculus*) | DBA/2 | Beijing Vital River Laboratory | Stock No.: 214 | |
| Genetic reagent (*M. musculus*) | PWK/PhJ | Jackson Laboratory | Stock No.: 003715 | |
| Gene (*Homo sapiens*) | *AAVS1* | GenBank | GeneID: 54776 | |
| Gene (*M. musculus*) | *Actb* | GenBank | GeneID: 11461 | |
| Gene (*M. musculus*) | *Sox2* | GenBank | GeneID: 20674 | |
| Gene (*M. musculus*) | *Rosa26* | GenBank | GeneID: 14910 | |
| Gene (*M. musculus*) | *Cdx2* | GenBank | GeneID: 12591 | |
| Strain, strain background (*Escherichia coli*) | Trans109 | TransGen Biotech | CD301 | Competent cells |
| Cell line (*Homo sapiens*) | HCA2-hTERT | *Gorbunova et al., 2002* (DOI: 10.1074/jbc. M202671200) | | Human fibroblast |
| Cell line (*Homo sapiens*) | HEK 293FT | ATCC | RRID:CVCL_6911 | |
| Cell line (*M. musculus*) | E14 | ATCC | RRID:CVCL_C320 | Mouse ES cell line |
| Cell line (*M. musculus*) | Oct4-ΔPE-GFP transgenic C57BL/6 × PWK ESC | This paper | | Mouse ES cell line |
| Antibody | Anti-RPA2 (Rabbit polyclonal) | Abclonal | A2189 RRID:AB_2764207 | WB (1:1000) |
| Antibody | Anti-NBS1 (Rabbit) | Cell Signaling Technology | 3002 RRID:AB_331499 | WB (1:1000) |
| Antibody | Anti-RAD50 (Rabbit polyclonal) | Abclonal | A3078 RRID:AB_2764881 | WB (1:1000) |
| Antibody | Anti-RAD51 (Mouse polyclonal) | Abcam | ab88572 RRID:AB_2042762 | IF (1:500), WB (1:1000) |
| Antibody | Anti-MRE11 (Rabbit polyclonal) | Abclonal | A2559 RRID:AB_2764447 | WB (1:1000) |
| Antibody | Anti-53BP1 (Rabbit polyclonal) | Cell Signaling Technology | 4937S RRID:AB_10694558 | IF (1:100), WB (1:1000) |
| Antibody | Anti-CtIP (Rabbit polyclonal) | Abcam | ab70163 RRID:AB_1209429 | WB (1:1000) |

*Continued on next page*

*Continued*

| Reagent type (species) or resource | Designation | Source or reference | Identifiers | Additional information |
|---|---|---|---|---|
| Antibody | Anti-BRCA1 (Rabbit polyclonal) | Abclonal | A0212 RRID:AB_2757026 | WB (1:1000) |
| Antibody | Anti-EXO1 (Rabbit polyclonal) | Abclonal | A6810 RRID:AB_2767391 | WB (1:1000) |
| Antibody | Anti-XRCC2 (Rabbit polyclonal) | Abclonal | A1800 RRID:AB_2763839 | WB (1:1000) |
| Antibody | Anti- TUBULIN (Rabbit polyclonal) | Abmart | M20005 | WB (1:2000) |
| Antibody | Anti-gamma H2AX (Rabbit polyclonal) | Cell Signaling Technology | 9718S RRID:AB_2118009 | IF (1:500) |
| Recombinant DNA reagent | sgRNA-Cas9 plasmid | PMID:28524166 | | Kind gift from Hui Yang lab |
| Commercial assay or kit | Click-iT EdU Assay Kit | Invitrogen | C10634 | |
| Commercial assay or kit | Comet assay | Trevigen | Cat. # 4250–050 K | |
| Chemical compound, drug | Farrerol | Sigma Aldrich | SML1389 | 0.05, 0.1, 1, 5, 10 µM |
| Chemical compound, drug | RS-1 | ApexBio | C3357 | 1, 7.5, 10 µM |
| Chemical compound, drug | SCR7 | Selleck | S7742 | 0.1, 1, 20 µM |
| Software, algorithm | ImageJ software | ImageJ (http://imagej.nih.gov/ij/) | RRID:SCR_003070 | |
| Software, algorithm | GraphPad Prism software | GraphPad Prism (http://graphpad.com) | RRID:SCR_015807 | Version 6.0 |
| Software, algorithm | CASPLab software | CASPLab http://casp.sourceforge.net | RRID:SCR_007249 | |
| Other | DAPI stain | Abcam | ab104939 | |

## Construction of plasmids

To create a doxycycline-inducible vector expressing the rare endonuclease, I-SceI, the open reading frame (ORF) of I-SceI enzyme was amplified, and inserted into the *AAVS1*-TRE-GFP-CAG-rtTA vector to replace the GFP gene using SalI and MluI digestion. A hygromycin resistant gene was also amplified and inserted to the vector at the NotI/KpnI site.

To generate an *AAVS1*-mCherry donor vector, the backbone was isolated from an *AAVS1* donor plasmid (*AAVS1*-SA-Puromycin-CAG-GFP) (*Mali et al., 2013*) by BglII digestion and gel purification. The p2A-mCherry insert was amplified from *Actb*-p2A-mCherry plasmid (*Yao et al., 2017*). Afterwards, the insert was recombined into the backbone using the ClonExpress MultiS Cloning Kit (Vazyme catalog no.C113).

To generate the puromycin-GFP donor for targeting the site of the mouse *Actb* gene, both homologous arms were amplified from the *Actb*-p2A-mCherry plasmid. They were then subcloned into the donor plasmid (*AAVS1*-SA-Puromycin-CAG-GFP) by replacing the two *AAVS1* homologous arms.

To create the p2A-BFP-WPRE donor for targeting the mouse *Rosa26* locus, the backbone was isolated from a *Rosa26*-GFP donor plasmid by MluI and PmeI digestion and gel purification. The insert

BFP-WPRE fragment was amplified from a pCMV-BFP plasmid. Finally, the insertion fragment was recombined into the backbone vector using the ClonExpress MultiS Cloning Kit (Vazyme catalog no. C113).

All the generated constructs were validated by Sanger sequencing.

## Generation of small molecule screening platform

The MscI linearized doxycycline-inducible I-SceI expression vector was transfected to the previously reported D4a cells (*Chen et al., 2019*) using the Lonza 4D electroporation machine with DT-130 program. At 24 hr post transfection, 30 µg/mL Hygromycin B was applied to the transfected cells for selection. On day 12, individual colonies were picked and expanded for further study.

## Cell culture and transfection

HCA2-hTERT fibroblasts and all derived cell lines were cultured in MEM (Sigma) supplemented with 10% FBS (Gibco, Cat. #: 10270–106),1% penicillin/streptomycin (Gibco, Cat. #: 15140–122) and 1% NEAA (Gibco, Cat. #: 11140–050). HEK 293FT cells were maintained in DMEM (Sigma, Cat. #: D6429) supplemented with 10% FBS (Gibco, Cat. #: 10270–106) and 1% penicillin/streptomycin (Gibco, Cat. #: 15140–122). All cells were cultured at 37 ˚C in a 5% $CO_2$ atmosphere.

Mouse ESCs (E14 cells and Oct4-ΔPE-GFP transgenic C57BL/6 × PWK ESC) were maintained in DMEM supplemented with 15% fetal bovine serum, 1% nonessential amino acid, 1% penicillin/streptomycin, 1% nucleosides, 1% L-glutamax, 0.1 mM mercaptoethanol, 1000 U/mL LIF, 1 µM PD0325901 (Selleck) and 3 µM CHIR99021 (Selleck).

HEK 293FT cells and mouse ESCs were transfected using the Lonza 4D electroporation machine according to the manufacturer's instructions. For HEK 293FT cells transfection, $2 \times 10^5$ cells were transfected with mixed plasmids (1 µg sg-*AAVS1* vector, 1 µg Cas9 expression plasmid and 1 µg donor plasmid) using CM130 program. For mouse ESCs, program of CG104 was used to transfect cells. Cells were treated with different small molecules for 24 hr before transfection. Farrerol was purchased from Sigma-Aldrich (SML1389). The concentration of small molecules used in cells were as follows: 0.1 µM or 1 µM SCR7 (Selleck) for HEK 293FT and mouse ESCs, and 1 µM or 20 µM SCR7 for mouse embryos. 1 µM or 10 µM RS-1 (ApexBio) for human and mouse cells, and 7.5 µM RS-1 for mouse embryos. 0.05 µM or 0.1 µM farrerol for mouse embryos, and 0.1 µM or 1 µM or 5 µM or 10 µM farrerol for human and mouse cells.

HEK 293FT cell line and mouse ESC E14 were obtained from ATCC. HCA2-hTERT cell line was obtained from Gorbunova lab (*Gorbunova et al., 2002*). The mouse ESC cell line used in karyotype analysis was derived from Oct4-ΔPE-GFP transgenic C57BL/6 (paternal) × PWK (maternal) blastocyst. All cell lines were routinely tested to ensure that they were mycoplasma free.

## Zygote injection, embryo culturing and generation of Knock-in mice

Zygote injection and culture was performed as previously reported (*Zheng et al., 2018*). Briefly, embryos were isolated from C57BL/6n or BDF1 female mice (6–8 weeks old). The female mice were super ovulated by intraperitoneally injecting with PMSG (5–6 IU) (Pregnant mare serum gonadotropin) and hCG (6–7 IU) (Human choionic gonadotophin), and then mated to C57BL/6n or BDF1 male mice. The zygotes were harvested from oviducts and received the injection of mixed mCherry-donor (100 ng/µL), Cas9 mRNA (100 ng/µL) and sgRNA (70 ng/µL) targeting *Actb* or *Cdx2* locus, then cultured in G1 plus medium (10128, Vitrolife) with small molecules for ~24 hr. Then the embryos were washed and cultured in new G1 plus medium without small molecules till blastocyst stage. In some cases, the zygotes were directly injected with small molecules and CRISPR/Cas9 mixture and cultured in G1 plus medium. The phenotype of embryos was evaluated on a fluorescent microscope and individual blastocysts were then picked for genotyping analysis.

For analysis of the toxicity of small molecules, the mouse zygotes were cultured in G1 plus medium containing individual small molecules till 2 cell stage or blastocyst stage. The *in vitro* developmental potential of these embryos was recorded until E4.5.

For generation of *Cdx2*-mCherry knock-in mice, zygotes were firstly injected with the combination of Cas9 mRNA (100 ng/µL), sgRNA targeting the *Cdx2* locus (70 ng/µL) and donor (HR donor or HMEJ donor, 100 ng/µL). Then the injected embryos were cultured in G1 plus medium (10128, Vitrolife) with or without farrerol (0.05 µM) for about 24 hr. Then, 15–20 2-cell-stage embryos were

transferred into the oviduct of a pseudopregnant female ICR mouse. The number of transferred embryos and born pups was recorded. The genotype of each offspring was verified through Sanger sequencing.

## Analysis of the genome targeting efficiency

For the analysis of gene targeting efficiency in HEK 293FT and mouse ESCs, the cells were electroporated with donors in the absence or presence of the indicated compounds. On day two post transfection, cells were treated with 1 μg/mL puromycin for approximately 1 week. Formed colonies were picked and lysed in 20 μL lysis buffer (0.45% NP-40, 2 mg/mL Proteinase K) for 1 hr at 56 ˚C and 10 min at 95 ˚C (*Li et al., 2017a*). The lysates were used as templates for PCR amplification.

For the analysis of gene targeting efficiency in mouse embryos, the procedure was performed as previously reported (*Yao et al., 2017*). Single embryos were transferred into PCR tubes containing 5 μL buffer G1 (25 mM NaOH, 0.2 mM EDTA, pH 12) and were lysed at 95 ˚C for 10 min. Then 5 μL buffer G2 (40 mM Tris-Cl, pH 5) was added to the PCR tubes for neutralization as the samples were cooled to the room temperature. The lysates were used as templates for nested PCR with two pairs of primers.

The PCR products were analyzed on 1% agarose gel and the expected bands were purified for sequencing.

## Production of Cas9 mRNA and sgRNA

The Cas9 mRNA and sgRNA were produced as previously reported (*Liu et al., 2017*). Briefly, Cas9 mRNA was *in vitro* transcribed using mMESSAGE mMACHINE T7 ultra transcription kit (Ambion, Thermo Fisher Scientific, USA). Meanwhile, sgRNAs were *in vitro* transcribed using MEGA shortscript T7 kit (Ambion, Thermo Fisher Scientific, USA). Both Cas9 mRNA and specific sgRNA were purified according to the standard protocol by phenol:chloroform extraction and ethanol precipitation, and then dissolved in DNase/RNase-free water (Life Technologies).

## Comet assay

Mouse ESCs were seeded at a density of $1 \times 10^5$ cells per well on 6-well plates. On day 2, cells were incubated with compounds at indicated concentrations. On day 3, cells were harvested for the analysis of genomic stability using alkaline comet assay (Trevigen, Gaithersburg, MD, USA, Cat. # 4250–050 K).

## EdU incorporation assay

Mouse ESCs at a concentration of $2 \times 10^5$ per well were seeded on 6-well plates, followed by compound treatment in 24 hr. After the treatment for another 24 hr, cells were incubated with 10 μM EdU for 2 hr at 37 ˚C. Cells were collected for conducting EdU assay followed by analysis on FACSverse using a Click-iT EdU Assay Kit (Invitrogen, Waltham, MA, USA, C10634).

## Immunofluorescence

For the immunofluorescence assay, cells cultured on coverslips were fixed with 4% paraformaldehyde for 15 min at room temperature, and washed with PBS for three times. Then the fixed cells were permeabilized with 0.25% Triton X-100 for 10 min, followed by 10 min-PBS washing for three times. Then the cells were blocked with 1% goat serum for 1 hr at room temperature, followed by the incubation with the indicated antibodies including anti-γH2AX (Cell signaling technology, Cat. #9718S), anti-53BP1 (Cell signaling technology, Cat. #4937S), anti-RAD51 (Abcam, Cat. #ab88572) or anti-RPA2 overnight at 4 ˚C. After washing with PBS for three times, the cells were incubated with the secondary antibody (Abcam, goat-anti-rabbit-FITC, Cat. #ab6717) for 1 hr at room temperature. Cells were then washed with PBS for three times before covered with mounting medium with DAPI (Abcam, Cat. #ab104939). The images were taken on a Nikon A1R laser scanning confocal microscope and analyzed with ImageJ.

## Karyotyping assay

The karyotype analysis was performed as previously reported (*Li et al., 2017b*). Briefly, one million Oct4-ΔPE-GFP transgenic C57BL/6 × PWK ESCs were firstly transfected with 3 μg sgRNA-Cas9

plasmid targeting the *Cdx2* locus and 2 µg donor plasmid. At 48 hr post transfection and treatment of indicated compounds, the cells were cultured in mouse ESC medium with 0.25 mg/mL colcemid (Invitrogen, Thermo Fisher Scientific) for 2 – 3 hr and collected with 0.05% Trypsin-EDTA (Invitrogen, Thermo Fisher Scientific). Then cells were incubated in hypotonic solution containing 0.4% sodium citrate and 0.4% potassium chloride at 37°C for 5 min, and were then fixed with a methanol/acetic acid mixture (3:1, v/v). The fixed cells were mounted on coverslips and stained with Giemsa at 37°C for 10–15 min after drying.

## Acknowledgements

We thank Dr. Hui Yang at Chinese Academy of Sciences for kindly providing Cas9-sgRNA vectors and donor vectors for knock-in at the *Actb*, *Cdx2*, *Sox2* and *Rosa26* loci. We thank Dr. Xiaoqing Zhang at Tongji University for kindly providing *AAVS1*-TRE-GFP-CAG-rtTA vector. We thank Dr. Michael Van Meter for critically reading the manuscript.

## Additional information

### Funding

| Funder | Grant reference number | Author |
|---|---|---|
| Chinese National Program on Key Basic Research Project | 2018YFC2000100 | Zhiyong Mao |
| Chinese National Program on Key Basic Research Project | 2017YFA0103300 | Ying Jiang |
| Chinese National Program on Key Basic Research Project | 2016YFA0100400 | Shaorong Gao |
| National Science Foundation of China | 31871438 | Zhiyong Mao |
| National Science Foundation of China | 81972457 | Ying Jiang |
| National Science Foundation of China | 31721003 | Shaorong Gao |
| National Science Foundation of China | 31871446 | Jiayu Chen |
| Fundamental Research Funds for the Central Universities | | Zhiyong Mao |
| Program of Shanghai Academic Research Leader | 19XD1403000 | Zhiyong Mao |
| "Shuguang Program" of Shanghai Education Development Foundation and Shanghai Municipal Education Commission | 19SG18 | Zhiyong Mao |
| The key project of the Science and Technology of Shanghai Municipality | 19JC1415300 | Shaorong Gao |
| Shanghai Rising-Star Program | 19QA1409600 | Jiayu Chen |
| Shanghai Municipal Medical and Health Discipline Construction Projects | 2017ZZ02015 | Xiaoping Wan |
| The Young Elite Scientist Sponsorship Program by CAST | 2018QNRC001 | Jiayu Chen |

The funders had no role in study design, data collection and interpretation, or the decision to submit the work for publication.

## Author contributions
Weina Zhang, Yu Chen, Data curation, Investigation, Writing - original draft, Writing - review and editing; Jiaqing Yang, Data curation, Writing - review and editing; Jing Zhang, Jiayu Yu, Data curation, Designed and conducted the experiments; Mengting Wang, Data curation; Xiaodong Zhao, Data curation, Provided helpful advice and feedback on various aspects of the study design; Ke Wei, Supervision; Xiaoping Wan, Supervision, Funding acquisition; Xiaojun Xu, Resources, Writing - review and editing; Ying Jiang, Data curation, Supervision, Funding acquisition, Writing - review and editing; Jiayu Chen, Data curation, Funding acquisition, Writing - original draft, Project administration, Writing - review and editing; Shaorong Gao, Supervision, Funding acquisition, Project administration, Writing - review and editing; Zhiyong Mao, Conceptualization, Supervision, Funding acquisition, Writing - original draft, Project administration, Writing - review and editing

## Author ORCIDs
Weina Zhang (i) https://orcid.org/0000-0003-2852-4079
Yu Chen (i) https://orcid.org/0000-0002-9661-3914
Jiayu Chen (i) https://orcid.org/0000-0003-4755-8901
Shaorong Gao (i) http://orcid.org/0000-0003-1041-3928
Zhiyong Mao (i) https://orcid.org/0000-0002-5298-1918

## Ethics
Animal experimentation: The specific-pathogen-free-grade mice, including C57BL/6n, ICR, DBA and BDF1 mice as well as generated knock-in mice, were housed in the animal facility at Tongji University. All the mice had free access to food and water. All the experiments were performed in accordance with the University of Health Guide for the Care and Use of Laboratory Animals, and were approved by the Biological Research Ethics Committee of Tongji University, and the approved protocol number was TJLAC-019-095.

## Decision letter and Author response
Decision letter https://doi.org/10.7554/eLife.56008.sa1
Author response https://doi.org/10.7554/eLife.56008.sa2

## Additional files

### Supplementary files
• Transparent reporting form

### Data availability
All data generated or analyzed during this study are included in the manuscript and supporting files. Source data files have been provided for Figures 1, 2, 3, 4, 5, 6 and figure supplements contained within 'Source data files'. Primer sequences named 'Table 1-source data 1' are also included as 'Source data files'.

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
