## [Decision Letter]

**Acceptance summary:**

Zhang et al. developed a screening platform using a dual fluorescent reporter for homologous recombination (HR) and non-homologous end joining (NHEJ) activity and report the identification of farrerol, a natural compound isolated from Rhododendron, that specifically enhances HR. The authors report a statistically significant increase in targeted integration using CRISPR-Cas9 mediated DSBs in HEK293 cells at the AAVS1 locus as well as in mouse embryonic stem cells at three different gene loci (*Actb*, *Rosa26*, *Sox2*). The compound does not lead to genotoxic stress as measured by γH2AX and 53BP1 foci and does not lead to karyotypic changes. Moreover, they show increased targeting activity in mouse embryos, which is statistically significant, an increase in germline transmission, and a control for absence of negative effect during embryonic development. The compound is readily commercially available and may substantially increase the efficiency of targeted integration in human and mouse cells, as well as aid in generating knock-in mice.

**Decision letter after peer review:**

Thank you for submitting your article "A high-throughput small molecule screen identifies farrerol as a potentiator of CRISPR/Cas9-mediated genome editing" for consideration by *eLife* as a Tools and Resources article. Your article has been reviewed by two peer reviewers, including Wolf-Dietrich Heyer as the Reviewing Editor and Reviewer #1, and the evaluation has been overseen by Jessica Tyler as the Senior Editor.

The reviewers have discussed the reviews with one another and the Reviewing Editor has drafted this decision to help you prepare a revised submission.

As the editors have judged that your manuscript is of interest, but as described below that additional experiments are required before it is published, we would like to draw your attention to changes in our revision policy that we have made in response to COVID-19 (https://elifesciences.org/articles/57162). First, because many researchers have temporarily lost access to the labs, we will give authors as much time as they need to submit revised manuscripts. We are also offering, if you choose, to post the manuscript to bioRxiv (if it is not already there) along with this decision letter and a formal designation that the manuscript is 'in revision at *eLife*'. Please let us know if you would like to pursue this option. (If your work is more suitable for medRxiv, you will need to post the preprint yourself, as the mechanisms for us to do so are still in development.)

Summary:

The manuscript describes the development of a screening platform using a dual fluorescent reporter on homologous recombination (HR) and non-homologous end joining (NHEJ) activity and the identification of farrerol, a natural compound isolated from Rhododendron, that appears to specifically enhance HR. The authors show that this compound enhances the HR signal by about 2.5-fold, whereas the NHEJ signal remains unchanged. They show a statistically significant increase in targeted integration using CRISPR-Cas9 mediated DSBs in HEK293 cells at the *AAVS1* locus as well as in mouse embryonic stem cells at three different gene loci (*Actb*, *Rosa26*, *Sox2*). Moreover, they show increased targeting activity in mouse embryos, which is statistically significant, an increase in germline transmission, and control for absence of negative effect during embryonic development. The compound is readily commercially available and may substantially increase to efficiency of targeted integration in human and mouse cells, as well as aid in generating knock-in mice.

Essential revisions:

1) The screening platform and the dual reported system is not explained well in Figure 1 and needs substantial clarification. In A, what does SA and SD is not defined. How long is the homology for the HR assay part? What is the control of I-SceI? rtTA, P_TRE_ and P_CAG_ are not defined.

2) The authors must discuss the unexpected result that HR is increased without a concomitant decrease in NHEJ. This is unexpected from the competition between both pathways, and typically mutants in one or the other pathway affect this balance in both directions. This result could also provide a key to understand the underlying mechanism, although it is understood that this contribution cannot address the underlying mechanism, but is focused on the observation as a tool.

3) The limited immunoblot analysis in Figure 1—figure supplement 2 does not allow to conclude that protein level changes are not causative, as only a small selection of HR proteins is analyzed and no negative regulators are considered. Excluding a mechanism by trying to demonstrate a negative from a small selection of targets is not a productive strategy, although it is useful to know that the protein analyzed show no changes.

4) The key issue that would farrerol make a useful tool is that it does induce genotoxic side effects, as shown for the SCR7 or RS-1. The analysis in Figure 4—figure supplement 2 tries to address this but needs to be expanded to include more markers of genotoxic stress (γH2AX foci, 53BP1 foci), karyotyping, and genomic analysis to monitor gross chromosomal rearrangements.

5) The germline transmission data and conclusion in Figure 5 are based on small numbers (5 versus 0 events), and it is surprising to see no germline transmission in the controls. Any explanation?

---

## [Author Response]

Essential revisions:1) The screening platform and the dual reported system is not explained well in Figure 1 and needs substantial clarification. In A, what does SA and SD is not defined. How long is the homology for the HR assay part? What is the control of I-SceI? rtTA, P_TRE_ and P_CAG_ are not defined.

Thank the reviewers for the suggestion. SD stands for splice donor, and SA stands for splice acceptor. The length of the homologous sequence in our dual reporter is ~ 3.9 kb including the Pem1 intron and the inserted full length-tdTomato. The rtTA is the reverse tetracycline-controlled transactivator; P_TRE_ is the promoter containing the Tet-responsive element, which can be activated upon rtTA binding to the sequence only in the presence of tetracycline or doxycycline; and the P_CAG_ is the CAG promoter, which is a strong synthetic promoter frequently used to drive gene expression in mammalian systems. We have included more clarification in figure legends (Figure 1; Figure 1—figure supplements 1 and 3). In our previous work (Chen et al., 2019), we characterized HCA2-D4a, the parental cell line of CLZ3 which we used for screening in this study, with a vector encoding the hypoxanthine phosphoribosyltransferase (HPRT) gene (pHPRT–CAG32) as the control of I-SceI. In HCA2-D4a cells transfected with the vector encoding HPRT gene, we did not observe GFP^+^ or tdTomato^+^ cells (Figure 1B, Chen et al., 2019), which we used as the measure of NHEJ and HR efficiency.

2) The authors must discuss the unexpected result that HR is increased without a concomitant decrease in NHEJ. This is unexpected from the competition between both pathways, and typically mutants in one or the other pathway affect this balance in both directions. This result could also provide a key to understand the underlying mechanism, although it is understood that this contribution cannot address the underlying mechanism, but is focused on the observation as a tool.

We thank the reviewers for the comments. We agree with the reviewer that HR and NHEJ are two competitive pathways when repairing DSBs. Nevertheless, upon the induction of a DSB in cells, several consequences might occur: (1) cells might choose not to repair it and enter senescence or apoptosis; (2) cells choose to repair it via canonical-NHEJ or alternative NHEJ; (3) cells choose to repair it by HR, which can be further categorized into two precise pathways, gene conversion and crossing over, and single-strand annealing (SSA) pathway. As shown in our paper, we also observed that farrerol actually suppressed SSA repair (Figure 1—figure supplement 3). Therefore, the reasons leading to the increase in HR with no concomitant decline in NHEJ might lie in that (1) the balance of three sub-HR pathways shifts from SSA to the two precise sub-HR pathways in the presence of farrerol; (2) fewer cells with the I-SceI induced DSBs enter senescence or apoptosis. As the reviewer suggested, we have included some discussion in the text.

3) The limited immunoblot analysis in Figure 1—figure supplement 2 does not allow to conclude that protein level changes are not causative, as only a small selection of HR proteins is analyzed and no negative regulators are considered. Excluding a mechanism by trying to demonstrate a negative from a small selection of targets is not a productive strategy, although it is useful to know that the protein analyzed show no changes.

We thank the reviewers for the suggestion. We agree with the reviewer on this issue. We have included more HR factors such as CtIP, EXO1, XRCC2 and the negative regulator such as 53BP1 in the immunoblot analysis. Our results showed that farrerol did not stimulate HR by influencing the expression of these proteins. This data is shown in Figure 1—figure supplement 2A. In addition, we have rephrased our text to tone down the conclusion.

4) The key issue that would farrerol make a useful tool is that it does induce genotoxic side effects, as shown for the SCR7 or RS-1. The analysis in Figure 4—figure supplement 2 tries to address this but needs to be expanded to include more markers of genotoxic stress (γH2AX foci, 53BP1 foci), karyotyping, and genomic analysis to monitor gross chromosomal rearrangements.

We thank the reviewers for the valuable suggestions. We have performed immunofluorescence experiments to investigate the level of more markers associated with genotoxicity, including γH2AX and 53BP1. The results showed that in consistent with the previous data obtained with the comet assay, in HCA2-hTERT fibroblasts farrerol treatment did not induce genotoxic side effects assayed by analysis of γH2AX and 53BP1 foci number (Figure 4; Figure 4—figure supplement 1). The γH2AX and 53BP1 immunofluorescence experiments were also carried out in mouse ESCs. By analyzing the fluorescence intensity of the γH2AX using the software ImageJ and the foci number of 53BP1, we observed that farrerol did not affect the genome integrity in mouse ESCs (Figure 4A-D). Moreover, we also performed karyotyping analysis to investigate the potential influence of these drugs on large-scaled chromosomal rearrangements. We found that both SCR7 and RS-1 treatment led to the increase in the percentage of cells with an abnormal karyotype while farrerol treatment had no obvious influence on the karyotype (Figure 4F). Taken together, these data strongly indicate that farrerol has no negative effect on genomic stability.

5) The germline transmission data and conclusion in Figure 5 are based on small numbers (5 versus 0 events), and it is surprising to see no germline transmission in the controls. Any explanation?

We thank the reviewers for the comment. In the previous version we attempted to emphasize that farrerol treatment can not only efficiently improve knock-in efficiency in embryos *in vitro* but also generate more founder mice *in vivo* compared with control, and the generated mice can further transmit the modified gene to the offspring. In the revised version, we performed more experiments by increasing the sample size in the experimental group with the farrerol treatment (Figure 6E; Figure 6—figure supplement 1C). And the data further supported the conclusion. However, the only one targeted founder mouse in the control group was unable to produce knock-in offspring, as shown in our previous version of manuscript. We agree with the reviewers that most founder mice have the ability to transmit their knock-in genotype to their offspring. Thus, we hypothesized that one possible reason for our observation is that the homologous recombination is inefficient in control group and there exists a mosaic genotype which is caused by the occurrence of gene editing even after the 2- or 4-cell stage in individual blastomere, leading to the generation of the chimeric mouse. And the cells with genomic locus not edited might contribute to PGCs (Primordial germ cells), leading to the disability of germline transmission we observed.